# Comparative Study of 2D Petrographic and 3D X-ray Tomography Investigations of Air Voids in Asphalt

**DOI:** 10.3390/ma16031272

**Published:** 2023-02-02

**Authors:** Moritz Middendorf, Cristin Umbach, Stefan Böhm, Jia Liu, Bernhard Middendorf

**Affiliations:** 1Institute of Transport Infrastructure Engineering, Technical University of Darmstadt, 64287 Darmstadt, Germany; 2Institute for Structural Engineering Department of Structural Materials and Construction Chemistry, University of Kassel, 34127 Kassel, Germany

**Keywords:** asphalt, void structure, imaging techniques, computer tomography, asphalt petrology

## Abstract

Knowledge of the exact composition of building materials (aggregate, binder, air voids, etc.) is essential for the further development of more resistant and sustainable building materials. In numerous scientific studies, the material behavior of asphalt is tested using mechanical methods. Here, the overall material behavior is determined (bitumen, air voids, aggregate). With the advent of imaging techniques, it is becoming possible to determine the individual constituents separately and perform a more detailed analysis of their location, shape and composition. Three-dimensional and two-dimensional methods are available for this purpose. For this study, two different types of asphalt (porous asphalt and asphalt concrete) were analyzed using 3D X-ray computed tomography and asphalt petrology as 2D methods; the results of both investigations are compared. The objective of this study is to determine whether the 2D method provides suitable results for the microstructural analysis of asphalt samples and how the results differ from those studied by the 3D method. The comparison shows that both methods can be used to analyze voids in asphalt samples. The 2D method provides valuable insight into the distribution of voids in a sample. In addition to the distribution of voids within a 2D section, the 2D method can also be used to make some structural statements about the location and structure of the voids in the 2D plane. The X-ray computed tomography method allows more complex analyses of the pore structure because of the third direction (3D). In addition, the 3D method provides more data, so that the pore structure can be described even more precisely, and the pore size (length, width, height) can be mapped and analyzed with a high degree of accuracy.

## 1. Introduction

With the advent of high-resolution imaging techniques in science, many research questions can be solved with this method in addition to mechanical testing. In structural road engineering, the material behavior of asphalt depends on the interaction between aggregates, bituminous binder and air voids. The aggregate must form a stable structure so that the traffic loads can be transferred in the subgrade. The bituminous binder binds the structure of the aggregate. The amount of binder defines if the asphalt structure is closed or open. Depending on the binder content and the aggregate particle size distribution, a distinction can be made between different types of asphalt. So-called asphalt concrete (AC) has a relatively dense structure with only a few voids due to the composition of bitumen and the grain size distribution of the aggregate. A porous asphalt (OPA) structure is high in void due to the large proportion of large grains [1]. Due to the strong dependence on the properties of bitumen at different temperatures, air voids (cavities) are needed in asphalt to relieve temperature-induced stresses. According to Krishnan and Rao [2], bleeding occurs when all voids are filled with bitumen. In this case, bitumen rises from the surface. Depending on their type, shape, size and cross-linking, the voids have different effects on the performance properties and durability of asphalt. For the characterization and quantification of the air void structure, imaging techniques are a suitable method, in addition to mechanical testing and dip weighing methods. In this context, investigations and scientific results are mainly based on 2D methods; therefore, thin or polished sections are examined.

First investigations with petrology for the characterization of asphalt were carried out in Denmark [3]. Later, the asphalt petrology method was introduced in Germany by Hart [4] and further developed by Tielmann and Hill [5] at the Technical University of Darmstadt. For this purpose, the voids are filled with colored resin for the purpose of analyzing them better. For example, in the work of Tielmann and Hill [5], different air void structures based on variations in compaction type and binder content were investigated. In addition, they visualized the interaction of bitumen and concerning crack propagation. 

Middendorf et al. [6] used asphalt petrology to analyze the effect of spray paver in asphalt composite structures. The layer boundary between the asphalt layers was observed based on a cross-section. Differences in the void structure were analyzed when the spray paver was used compared to conventional paving.

Considering the complexity of asphalt compositions, the accuracy of 2D methods may vary. Two-dimensional methods determining the void structure can lead to asphalt samples being considered similar in terms of their air void properties, even though their air void structures are different (e.g., in [7,8]). Using the 3D X-ray computed tomography (CT) technique, it is possible to capture the complete structure; it also allows a better understanding of the microstructure morphology. The application requires the development of volumetric quantification methods for this purpose [9].

Partl [10] conducted the first investigations using computed tomography on asphalt specimens in 1973. Using mastic asphalt specimens, void inclusions with diameters up to approximately 2.0 mm were detected. In 1999, Braz et al. [11] demonstrated the formation of cracks in asphalt specimens after different loading cycles using computed tomography.

In particular, the 3D structure of porous asphalt has been investigated in numerous research works [1,12,13,14]. For porous asphalts, the void structure is fundamental, as this is the most common solution for noise reduction and roads with a minimized risk of aquaplaning after heavy rainfall [15]. Accurate knowledge about the void structure is required to determine the material behavior [16]. The problem with porous asphalts is, on the one hand, the abrasion of the binder due to drainage and, on the other hand, the sealing with dirt. In their research work in 2018, Alber et al. [12,13] examined dirty and clean specimens of porous asphalt with CT and showed correlations with sound absorption. In addition to analyzing void structures, Chen et al. [17] used X-ray computed tomography to characterize fatigue damage in asphalt mixtures.

The computed tomography method is not exclusively used in science to observe void structurers. Middendorf et al. [18] developed an analysis method for viewing the layer boundary of asphalt structures using CT. By adding tracers with high density, specific subareas in the asphalt specimen can be highlighted and viewed more closely with imaging. The procedure has mainly been used for analyzing bitumen emulsion at the layer boundary (Figure 1).

In this study, the void structure of two different types of asphalt (porous asphalt and asphalt concrete) are analyzed using two different imaging techniques. For this purpose, on the one hand, the 2D method of asphalt petrology is used; on the other hand, the 3D method of X-ray computed tomography is used. The measurements are carried out on identical specimens so that the results can subsequently be compared.

As described in the literature study, there are many findings regarding the 3D method (computed tomography) for the analysis of the void structure. Conventionally, this is performed for asphalt samples with a weighing method, which is, however, very error-prone and does not allow a description of the void structure, but only provides the void content as a value.

For this reason, the aim of the study was to determine whether the 2D method provides suitable information for the microstructure analysis of asphalt samples, and whether it is comparable to the already recognized method of computed tomography.

## 2. Materials and Methods

### 2.1. Asphalt Specimen

This study used two different asphalt types to compare the two imaging techniques. A porous asphalt (OPA) is characterized by a high percentage of voids in the structure and a coarse-graded grain size distribution with a small number of fines. The void structure should be designed to allow water to pass through the asphalt layer. Porous asphalts are used on highways to prevent aquaplaning and spraying, and because of the pavement’s noise-reducing properties [19]. In addition, asphalt concrete is investigated, which is a heterogeneous asphalt consisting of voids, bitumen and aggregates [20]. Asphalt concretes as a surface layer material usually have a void content of 5.0 to 7.0 vol.-% and consist of a graded particle size. The particle size distributions of the two asphalts are shown in Figure 2.

In addition to the particle size distribution, the two asphalts differ in their void content, binder content and the used aggregate types. The differences are shown in Table 1. The void content and the void structure are particularly relevant for the evaluation with imaging methods.

Marshall specimens were produced from the two asphalts to compare the imaging methods. Specimens with dimensions of 10.0 × 4.0 × 6.3 cm were cut out of the Marshall specimens. In the first step of the study, the specimens were measured with high-resolution X-ray computed tomography (µ-CT). The second step used identical specimens for the 2D asphalt petrology analyses. Three specimens were analyzed for each asphalt type.

### 2.2. Two-Dimensional Method—Asphalt Petrology

The 2D imaging technique used in this study is asphalt petrology. A special coordinated preparation technique carries out the preparation of the specimens in asphalt petrology. In this process, sections are prepared from drill cores or other samples (e.g., Marshall specimens and plates from the rolling sector compactor). The ground sections should have a width of approximately 100 mm to 150 mm so that a representative sample of the structure can be imaged. A cut-off saw with diamond-tipped saw blades is used to cut the ground sections from the specimens. Since the binder tends to soften during cutting and, thus, smears, water must be used as a cooling agent during the cutting process (see Figure 3). After cutting, the specimen must be dried. When the specimen is completely dry, vacuum impregnation in a vacuum chamber at a minimum pressure of 3 kPa with colored and fluorescent epoxy resin will fill the air voids. This causes the air voids to become prominent and visible against the rock and the binder. Protruding epoxy resin is ground off after complete curing (see Figure 4), so it is only present in the filled voids.

The prepared sections are then recorded with a flatbed scanner. The generated UV scan can then be analyzed with evaluation programs. In this study, the evaluation was performed with the program JMicroVison (freeware, Vision 1.3.3). JMicroVison is specifically designed for the analysis of rock thin sections and allows the classification and quantification of different components. All individual cavities are read out and stored in tabular form using digital image processing. The data that can then be determined are, for example, the absolute void content, the vertical and horizontal void distribution, and the void orientation [21]. The different data types are listed in Table 2.

The ratio of the number of air void pixels to the total number of pixels in the sample calculates the total air void content. According to Equation (1), this is determined:(1)$$V_{air\;void}=\frac{\sum pixel\;air\;void}{\sum pixel\;picture}$$

The horizontal and vertical void distribution are calculated similarly. By segmenting the image, an image contains only the following information: Void = 1 or aggregate/binder = 0. A graphical representation can be formed from this matrix if this information is transferred to a matrix. This is schematically shown in Figure 5 and calculated in Equation (2).
(2)$$V_{air\;void,vertical,n}=\frac{\sum \prime \prime 1\prime \prime }{\sum \prime \prime 1\prime \prime +\sum \prime \prime 0\prime \prime }$$

### 2.3. Three-Dimensional Method—X-ray Computed Tomography

The increasing use of imaging techniques in materials research leads to the wish for 3D images. X-ray computed tomography (CT), which was first used in medicine, is becoming increasingly important. CT imaging can generate three-dimensional images of material and analyze the microstructure based on density differences. To be able to image the fine and very complex structures of the materials, X-ray microscopy is used for this purpose. First, magnification can be achieved by the position of the sample between the detector and the X-ray source. Here, the beam set can be used via the distance from source to sample and sample to the detector. This is also called geometric magnification. Second, detector units with scintillation counter convert the X-rays into visible light, which makes magnification via lenses possible. A CCD camera (charge-coupled device) records the magnified X-ray images. To generate a 3D image, X-ray images are taken at different angles and combined to form a three-dimensional data set. The calculation is performed by a filtered back projection and is also called reconstruction. The procedure of imaging with a 3D X-ray microscope is shown schematically in Figure 6.

Resolutions of up to 0.7 µm can be achieved with high-resolution X-ray computed tomography (µ-CT). For a complete 3D reconstruction of a sample, images are captured in an angular range of at least 180°. To avoid artifacts, the samples are rotated 360° during the measurement. The individual images created during the measurement have different gray values depending on the density of the material (aggregate, bitumen, air). As the images are recorded in 16 bits, the gray value ranges from 0 to 65,535. The gray values depend on the radiation absorption of the materials. A high absorption results in a low signal on the detector. If the sample is more permeable to radiation, the signal will be more intensive. The radiation absorption of the sample depends on factors such as density, atomic number and material thickness. Materials with a high density would consequently result in a dark area. This makes it difficult to understand the images because X-ray images are always displayed inverted. This means that dense structures are displayed brighter and less dense structures darker. CT images are also inverted for this reason.

In this study, high-resolution X-ray computed tomography (µ-CT) from Carl Zeiss Microscopy GmbH (Oberkochen, Germany, type: Xradia 520 Versa) was used (Figure 7). The X-ray computed tomography can measure with a voltage range of 30 to 160 kV and a power of 1 to 10 W. A spatial resolution of ≥0.7 µm can be achieved. The 12 filters (6 low-emission and six high-emission filters) and the surface detector (magnification factor 0.4×) or the three different objectives (4×, 20× and 40×) allow the different properties of specimens to be suitably recorded. The measurement parameters of the µ-CT acquisitions are summarized in Table 3.

Reconstruction of the recorded projections is performed automatically or manually using the XMReconstructor software. The evaluation of the 3D reconstruction of the samples is then carried out using the Avizo 9.4 software from Thermo Fisher Scientific Inc. (Waltham, MA, USA). Avizo is a material characterization and quality control software for image data analysis. Precise segmentations are possible with the software, allowing models to be created which can then be mathematically evaluated. Through this procedure, geometric quantities, position parameters, histograms or the aspect ratio of the materials can be read out and provide information about the microstructure of the specimens.

The data in Table 4 were extracted from the created reconstructions and void models of the different asphalt types for the study.

### 2.4. Comparison Parameters: 2D and 3D

For the comparison between the 3D method of X-ray computed tomography and the 2D method of asphalt petrology, the classical measured values, such as void content, void size and pore number, are of interest, in addition to the presented measurement parameters. Table 5 shows the measurement parameters used for the comparison.

## 3. Comparison of the Results between 2D and 3D Methods

The comparison of the two imaging methods was carried out on identical specimens that were also used for 3D investigations. In the first step, the specimen was examined non-destructively using the µ-CT method and then re-examined using the asphalt petrology method. This procedure allows the results to be directly compared with each other.

Using the 2D asphalt petrology method, sections of the porous asphalt were prepared and then analyzed. Figure 8 shows an example of a typical image of porous asphalt produced by the asphalt petrology process. The voids are filled with fluorescent epoxy resin to be analyzed in the recorded scan. Figure 8 shows 136 voids with an area fraction of 15.25%. In addition to the area fraction of the voids, the distribution of the voids in the 2D image can also be analyzed. Figure 8 shows the void distribution in the vertical direction. Based on the void distribution, a relatively uniform distribution of the voids can be seen. In addition, the cross-section view suggests that the specimen consists of individual larger voids (≤120.5 mm^2^).

Figure 9 shows the visualized 3D void structure of the µ-CT measurement of the porous asphalt (OPA). The specimen is identical to the one in Figure 8 and shows a total of 2059 voids in a volume of 6480 mm^3^. A closer look at the three-dimensional void structure in Figure 9 shows that the voids are interconnected and form a large void. This is particularly evident from the color representation of the cavities. The large void with a volume of 6202.56 mm^3^ is shown in blue. In addition, many smaller voids (yellow, green, orange and red) can be seen within the specimen. Unlike assumed in the 2D method, the specimen does not consist of many large cavities (area up to ≤120.5 mm^2^) but most of one large void (volume 6202.56 mm^3^). The large void represents 95.60% of the total void volume.

In addition to porous asphalt, asphalt concrete was also investigated in this study. Figure 10 shows an example of a cross-section of the asphalt concrete investigated. Based on the 2D section, many small voids (≤30.13 mm^2^) can be identified. The cross-section shown in Figure 10 shows 1079 voids with a total area of 7.71%.

Figure 11 shows the 3D void structure of the same specimen as in Figure 10. A total of 6611 air voids with a volume of 1688 mm^3^ are shown. As already analyzed in Figure 9, the voids are interconnected. The volume of the largest void (shown in blue) is 335 mm^3^ and represents 19.84% of the total void volume.

In the two-dimensional analysis, an average porosity of 14.56% with a mean data volume of 273 air voids was determined for all samples of the porous asphalt. The 3D data imply a mean total porosity volume of porous asphalt (OPA) of 29.13% with 2086 air voids. For the asphalt concrete (AC), the 2D method measured an average of 751 air voids and a total porosity of 6.58% for all specimens. The 3D method analyzed a total porosity of 17.05% with 5554 air voids. The lower limit was set to 0.0049 mm^2^ for the 2D method and 0.00025 mm^3^ for the 3D method. Table 6 compares the described results of the two methods and the two asphalts.

A comparison of the results shows that the 2D analysis produces the porous asphalt at approximately less than 13% of the data volume of a similar 3D analysis, and for the asphalt concrete, less than 15% of the data volume. This shows that the asphalt petrology method reveals only part of the void structure of the specimen. The voids within a 2D image are considered individual voids, which do not provide information about the entire structure. To obtain information about the specimen’s whole void and void structure, tomography is necessary.

In addition to the classic parameters, such as void size, several voids and porosity, imaging techniques can be used to analyze more factors that describe the shape of the voids. For example, the shape and orientation of the voids influence the mechanical behavior of asphalt, water permeability and noise reduction in porous asphalts. Thus, it has been demonstrated that there is a shift in the frequency of the absorption maximum when voids are arranged in the deep region of the layer compared to a homogeneous distribution of voids [23].

Figure 12 compares the measured length and width of the voids between the asphalt petrology method (2D) and the X-ray computed tomography method (3D). The regression line is compared in this diagram. The comparison of the straight lines in the left diagram for the asphalt concrete shows that the two methods give relatively identical results. The coefficient of determination R^2^ for the 3D method for the measured specimens is between 0.94 and 0.96. For the 2D method, the coefficient of determination R^2^ is between 0.85 and 0.91. Similar results were observed with the porous asphalt. The coefficient of determination R^2^ is between 0.95 and 0.97 for the 3D measurement and between 0.89 and 0.93 for the 2D method.

Figure 12 shows that the aspect ratio is slightly overestimated with the 2D method compared to the 3D method. In addition, the comparison of lengths and widths shows that significantly larger lengths (+395% for AC, +256% for OPA) and widths (+316% for AC, +216% for OPA) are measured with the 3D method. This is because of the more significant amount of data generated in a complete 3D acquisition. Nevertheless, the 2D method is a suitable indicator for estimating the aspect ratio.

In addition to the length and width of the voids, the height of the voids was also compared. The height of the air voids was compared with the volume/area in Figure 13. For this purpose, the parameters were shown cumulatively.

The left diagram shows the results of the asphalt concrete specimens of the 2D method and the 3D method. It can be seen from the diagram that the 2D method analyzes small heights of air voids. The plotted lines show an almost linear course. In contrast, the curves of the lines from the results of the 3D method show step changes from a certain height (>35 mm) up to the maximum volume (100%). The step of the course represents the large voids. As the voids within a specimen are interconnected, this single large void has the greatest volume of the specimen.

This finding becomes even more transparent for the open-pored specimens. For example, for specimen OPA1, the single large void with an average height of 46 mm accounts for 96.10% of the complete volume.

The heights and volume/area comparison shows that the asphalt petrology (2D) method underestimates the heights. Considering the voids as single voids means they are analyzed as too small in terms of average height.

Third, the measured elongation was compared between the 2D and 3D method. Elongation describes the ratio between length and width, and it can be calculated using the following formula.
(3)$$\mathrm{Elongation}=\frac{\mathrm{Wide}}{\mathrm{Length}}$$

In this notation, the elongation can assume a range of values between 0 and 1. A value of 1 means that the shape is round, and smaller values describe increasingly elongated voids.

Figure 14 shows the comparison of the elongation between the 2D method and the 3D method for both asphalts. The left diagram shows the distribution of elongation for asphalt concrete and the right plot shows porous asphalt.

In the left diagram (asphalt concrete), the distribution of the 2D measurement is in the range between 0.1 and 0.4, indicating relatively smaller rounding. The elongation for the 3D measurement is instead in the higher range ≥ 0.4. With a relative frequency of 11%, the peak elongation is about 0.6.

In the right diagram for the porous asphalt, the distribution of elongation for the 3D measurement is more uniform. Here, the values between 0.1 and 0.7 are on an equal level. The distribution for the 2D measurement shows a clear peak elongation with a relative frequency of 14% at 0.5.

Bacaicoa et al. [9] also compared the elongation of a 3D method and a 2D method in their study. Differences between the results were also detected. These differences were explained by the fact that the third axis is not considered in the 2D analyses, and the amount of data is smaller. This study found differences for all the tested specimens (porous asphalt, asphalt concrete).

## 4. Conclusions

This study compared two imaging techniques (2D and 3D). Currently, the two methods are the only ones that allow a structural void analysis of asphalt specimens. The objective was to determine to what extent the results are comparable and what indications the asphalt petrology method (2D) gives on the void structure of asphalt specimens. The study showed that the 2D method of asphalt petrology could provide insights into the voids of asphalt samples. The method provides valuable insights into the distribution of voids within a sample. For example, the asphalt petrology method can analyze whether the void distribution in the 2D plane is uniform or non-uniform. In addition to the distribution of voids within a 2D section, the 2D method can be used to make some structural statements in the 2D plane about the location and structure of the voids. Comparing the two imaging methods (asphalt petrology and X-ray computed tomography) showed that the aspect ratio can be estimated well with the 2D method. The coefficient of determination R^2^ for the 3D method ranged from 0.94 to 0.97 for the measured samples, and for the 2D method, the coefficient of determination R^2^ ranged from 0.85 to 0.91.

The X-ray computed tomography method makes more complex analyses of the void structure possible due to the third direction (3D). The voids within a sample are analyzed in 3D space, allowing the analysis of the connectivity of the voids. In addition, the 3D method provides more data, allowing the void structure to be described even more accurately and the void size (length, width, height) to be mapped and analyzed with a high degree of precision. This is particularly necessary for a complete scientific understanding of the material.

Asphalt petrology provides much smaller data so that an analysis can be performed quickly. Initial findings are available after only a few analysis steps using open source software. A complex analysis with specialized software is necessary for the X-ray computer tomography method.

The comparison showed that both methods could analyze voids in asphalt samples. The 2D method of asphalt petrology is swift and cost-effective for this purpose and already provides initial findings on the pore structure based on 2D sections. In addition, very large samples can be analyzed quickly and easily. The results of the 2D method can be used efficiently for a simulated assessment of the pore structure of uniformity and aspect ratio. Compared to asphalt petrology, the X-ray tomography method is expensive and only available in some places. However, the method is more accurate, shows more voids, and thus, provides a better scientific understanding of the void structure of asphalt samples. In summary, the user must decide for what purpose the investigation of the void structure is necessary. Asphalt petrology is an excellent and fast method for the standard examination and initial assessment of the void structure of asphalt specimens. For a complete void analysis and an accurate understanding of the complexity of the voids (connectivity, height, width), the 3D method of X-ray computed tomography is necessary.

## Figures and Tables

**Figure 1 materials-16-01272-f001:**
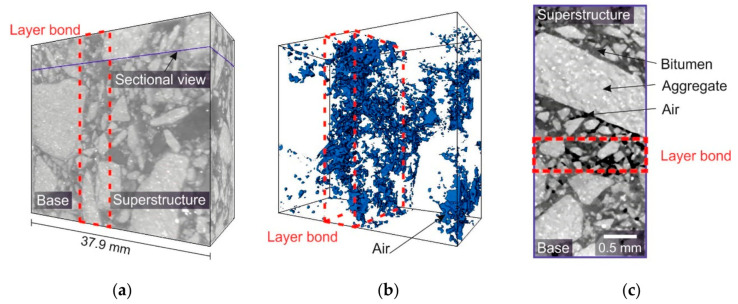
CT-scan of an asphalt composite sample with a resolution of 42.9 µm: (**a**) 3D reconstruction of the sample which provides an overview of the measurement; (**b**) model of the entrapped air; (**c**) sectional view into the sample [18].

**Figure 2 materials-16-01272-f002:**
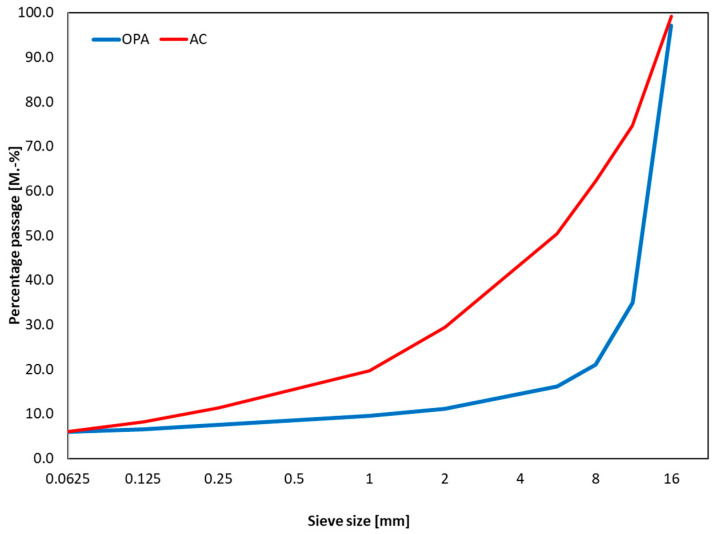
Aggregate gradation; asphalt concrete = red line; porous asphalt = blue line.

**Figure 3 materials-16-01272-f003:**
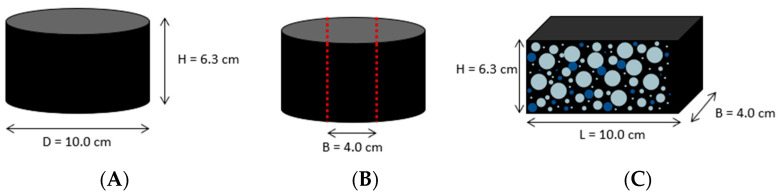
Specimen preparation for the measurements with the imaging techniques ((**A**) = specimen; (**B**) = cut area for preparation; (**C**) = cut specimen).

**Figure 4 materials-16-01272-f004:**
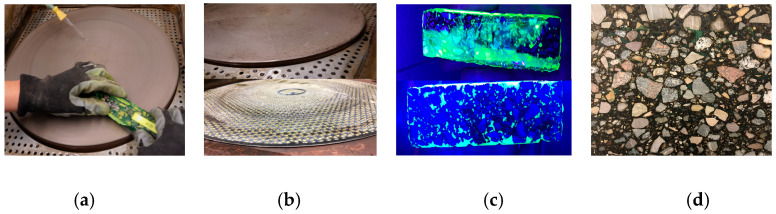
The grinding process of asphalt petrology: (**a**) water-cooled grinder; (**b**) different grinding wheels; (**c**) specimen before and after grinding process; (**d**) surface of specimen after grinding process (epoxy resin is only left in the voids) [18].

**Figure 5 materials-16-01272-f005:**
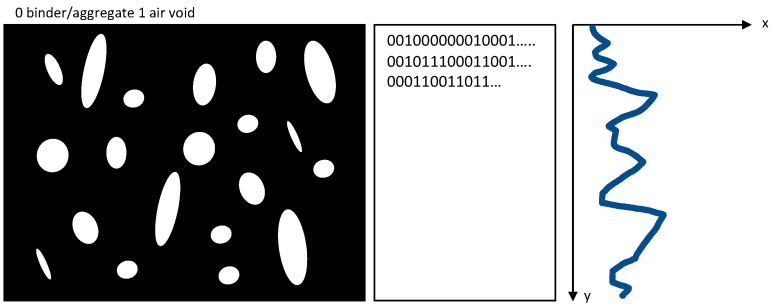
Creation of vertical and horizontal air void profiles in accordance with [21].

**Figure 6 materials-16-01272-f006:**
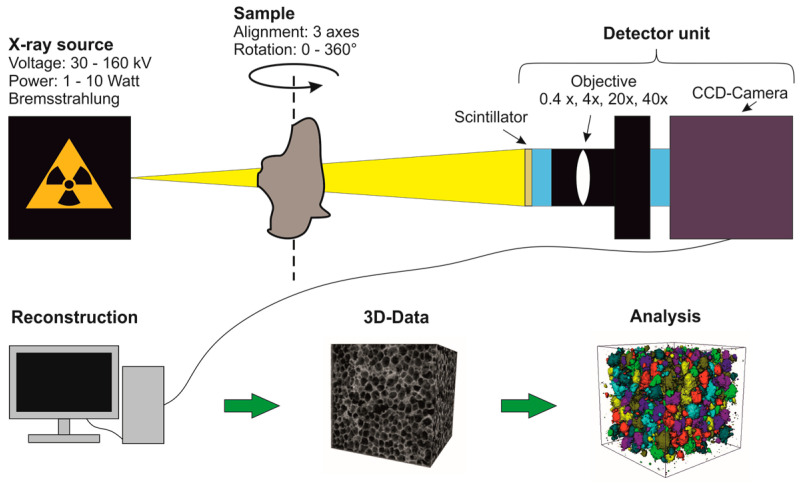
Functionality of the measurement and preparation of high-resolution X-ray computed tomography [22].

**Figure 7 materials-16-01272-f007:**
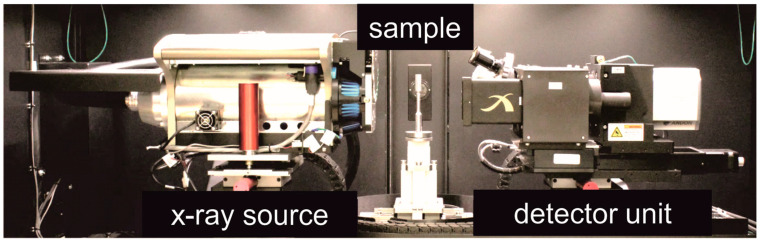
High-resolution computed tomography; type: Xradia 520 Versa [22].

**Figure 8 materials-16-01272-f008:**
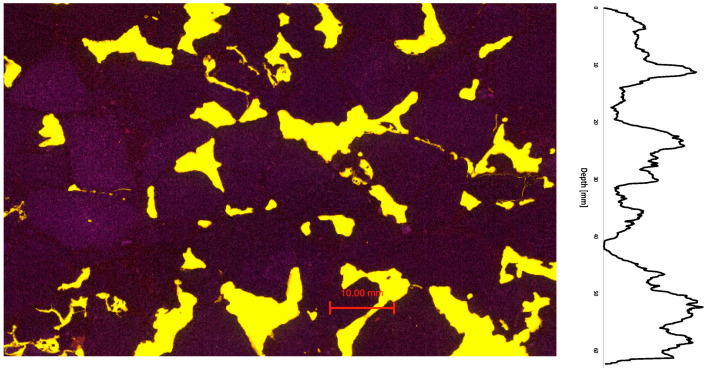
Grinding of porous asphalt (OPA)—asphalt petrology. Recorded scan (**left**) and vertical void distribution (**right**); area: 5.587 mm^2^.

**Figure 9 materials-16-01272-f009:**
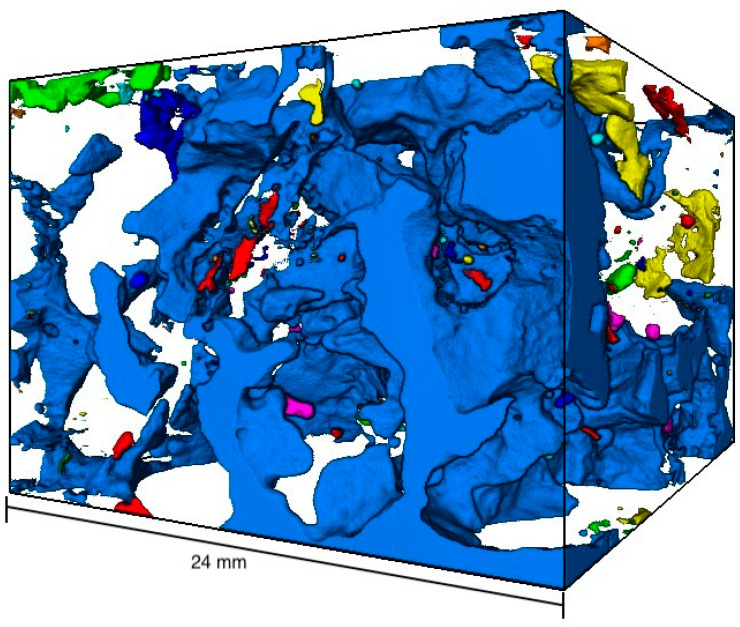
Visualized void distribution of the µ-CT scan—porous asphalt; volume: 48,593.1 mm^3^.

**Figure 10 materials-16-01272-f010:**
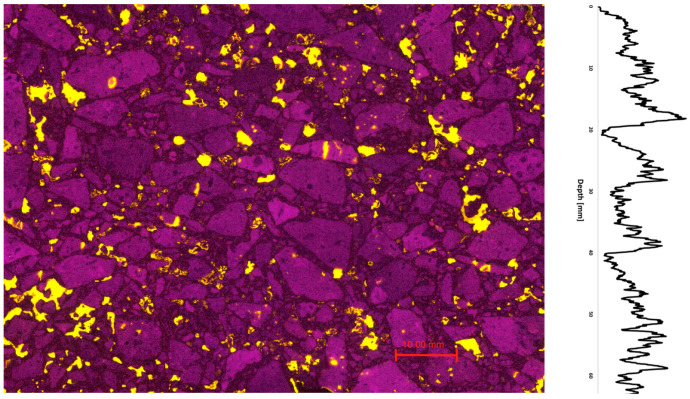
Grinding of asphalt concrete (AC)—asphalt petrology. Recorded scan (**left**) and vertical void distribution (**right**); area: 6.044 mm^2^.

**Figure 11 materials-16-01272-f011:**
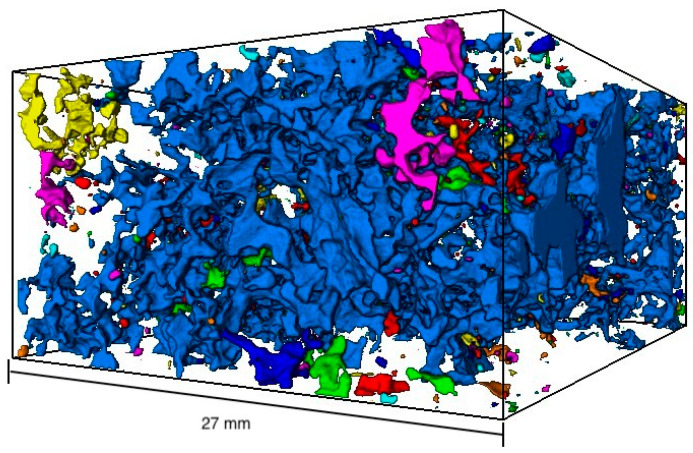
Visualized void distribution of the µ-CT scan—asphalt concrete; volume: 43,173.6 mm^3^.

**Figure 12 materials-16-01272-f012:**
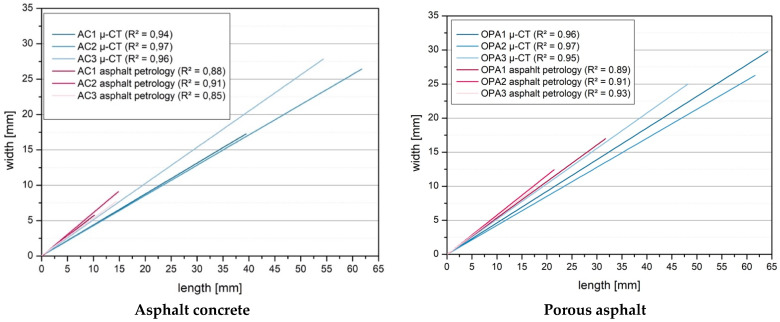
Comparison between length and width (2D method (asphalt petrology) and 3D method (µ-CT)).

**Figure 13 materials-16-01272-f013:**
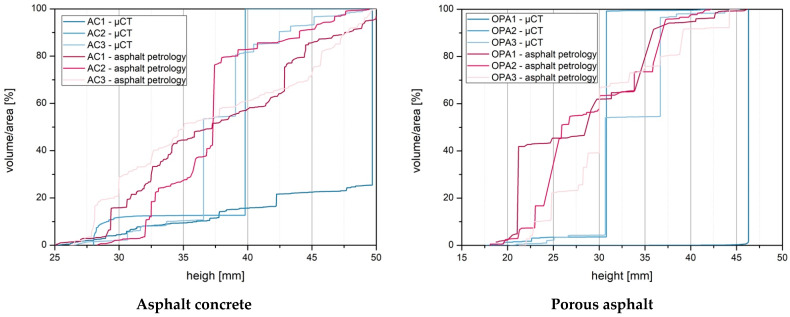
Comparison of height and volume/area (2D method (asphalt petrology) and 3D method (µ-CT)).

**Figure 14 materials-16-01272-f014:**
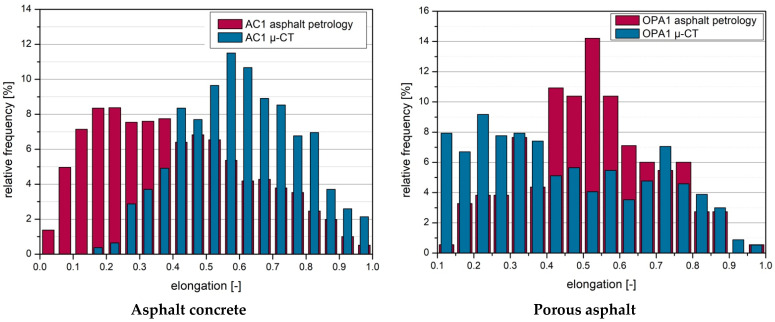
Comparison of elongation (2D method (asphalt petrology) and 3D method (µ-CT)).

**Table 1 materials-16-01272-t001:** Mixture types.

Mixture Property	Asphalt Concrete	Porous Asphalt
Asphalt binder content [M.-%]	4.4	5.4
Air void content [M.-%]	4.5	24.0
Aggregate type	rhyolite	basalt

**Table 2 materials-16-01272-t002:** Data types from the 2D evaluation.

Type	Details
Area	Expansion of the cut air void area
Perimeter	Length of the boundary
Length	Longest expansion along the major axis
Orientation	Angle between the principal axis of an air void and the horizontal axis
Width	Longest expansion along the minor axis
Equivalent circular diameter	Diameter of a circle with the same area
Feret diameter	Longest expansion along defined orientation
Coordinates of the barycenter or centroid	Arithmetic mean position of the air void pixels

**Table 3 materials-16-01272-t003:** Measurement parameters.

Measurement Parameter	
Source setting	140 kV; 10 W
Source filter	HE 3 (copper)
Source distance	−100 mm
Detector distance	100 mm
Resolution	34.6 µm
Optical magnification	0.4×
Exposure time	6 s
Physical size (diameter/ height)	65.1 mm/34.5 mm

**Table 4 materials-16-01272-t004:** Data types from the evaluation 3D.

Type	Details
Volume3d	Volume of the object.
Length3d	Length3D is the longest Feret diameter.
Width3d	Width3D is the shortest Feret diameter.
Area3d	Area of the object boundary.
Elongation	The ratio of the medium to the largest eigenvalue of the covariance matrix. Elongated objects will have small values close to 0.
OrientationPhi	Phi orientation of the particle in degrees [0, +90], computed with the inertia moments.

**Table 5 materials-16-01272-t005:** Two-dimensional and three-dimensional comparison parameters.

Parameter	Unit	Measurement Method
Void content	[%]	2D and 3D
Number of pores	[-]	2D and 3D
Volume	[mm^3^]	3D.
Area	[mm^2^]	2D
Elongation	[-]	2D and 3D
Length	[mm]	2D and 3D
Width	[mm]	2D and 3D
Height	[mm]	2D and 3D

**Table 6 materials-16-01272-t006:** Results of the 2D method and the 3D method.

	Porous Asphalt		Asphalt Concrete	
Properties	2D	3D	2D	3D
Mean numbers of voids [-]	273	2086	751	5554
Standard deviation of voids [-]	130	612	246	1481
Max. area; volume [mm^2^; mm^3^]	619.75	4827.31	503.41	1593.48

## Data Availability

The data presented in this study are available on request from the corresponding author.

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
