# Peer review of "Comparative Study of 2D Petrographic and 3D X-ray Tomography Investigations of Air Voids in Asphalt"

_materials, 2023, doi:10.3390/ma16031272_

Round 1

Reviewer 1 Report

The article proposes a method to study air voids in asphalt. It is possible to compare the difference in the unit weight of each specimen between asphalt concrete and porous asphalt.

Author Response

Dear Reviewer,

Thank you very much for your constructive comments on our paper. We have revised the paper accordingly and tried to fulfill the comments.

In the following, we will briefly explain how we have incorporated your comments and where you can find the changes in the paper.

The article proposes a method to study air voids in asphalt. It is possible to compare the difference in the unit weight of each specimen between asphalt concrete and porous asphalt.

The conventional method for determining the void content of asphalt samples is by weighing the samples in the dry state, under water and after storage in a water bath. However, this method is highly error-prone and even simple deviations in the results of weighing, lead to a different void content. For this reason, we have been looking for methods to achieve a better and more reliable result. The two selected methods are presented and compared in the paper.

We have revised the abstract so that it fits better to the content of the paper and gives a short overview of the paper. The changes can be seen in the edited submission (line 11-29).

Reviewer 2 Report

1.Please need to add legend for some figures related to the the size and area of voids.

2.For the software used need more details.

3.Need to see and consider the remarks marked on the research text. 

Author Response

Dear Reviewer,

Thank you very much for your constructive comments on our paper. We have revised the paper accordingly and tried to fulfill the comments.

In the following, we will briefly explain how we have incorporated your comments and where you can find the changes in the paper.

1.Please need to add legend for some figures related to the the size and area of voids.

We have included the volume and area information in the description of Figures 8-11.

2.For the software used need more details.

We have added some more information about the software. See line 153-154 and 219.

3.Need to see and consider the remarks marked on the research text. 

We have taken out the information with the chapter in each case.

In line 253 and 258, you wanted to know how we determine the following results.

That is explained in line 148 to 165. We can read out the yellow areas with the JMicroVision program. To do this, we set a threshold value that corresponds to the yellow area. With the program then all yellow areas are read out. The yellow areas correspond to the voids. We can then output the results as an Excel file and see the parameters from Table 2 for each yellow area.

This allows us to analyze the size and number of voids.

Reviewer 3 Report

Dear Author. Congratulations on the work you have done.

This paper uses 2D and 3D methods to characterize pores in asphalt concrete.

The introduction is at a sufficient level, it explains all the necessary information for the next parts. Materials and Methods. In this section, some paragraphs can be improved. Results and discussion. The text part is sufficiently exhaustive. Conclusion. The conclusion section briefly summary the findings of the study. The list of used publications is sufficient.

In general the structure and content of the manuscript is acceptable for materials. Nevertheless, in order to improve its readability, please consider the following suggestions.

1:The object of the article is the pore space in asphalt concrete, it is suggested that the title be changed from asphalt to asphalt concrete

2:The abstract includes too much background information and lacks clear conclusions.

3:The citation format of the references in the text should be revised according to the requirements of the journal

4:What are the technical advantages of using 2D petrographic, given that CT is also capable of obtaining 2D individual sections of concrete?

Author Response

Dear Reviewer,

Thank you very much for your constructive comments on our paper. We have revised the paper accordingly and tried to fulfill the comments.

In the following, we will briefly explain how we have incorporated your comments and where you can find the changes in the paper.

Dear Author. Congratulations on the work you have done.

This paper uses 2D and 3D methods to characterize pores in asphalt concrete.

The introduction is at a sufficient level, it explains all the necessary information for the next parts. Materials and Methods. In this section, some paragraphs can be improved. Results and discussion. The text part is sufficiently exhaustive. Conclusion. The conclusion section briefly summary the findings of the study. The list of used publications is sufficient.

In general the structure and content of the manuscript is acceptable for materials. Nevertheless, in order to improve its readability, please consider the following suggestions.

1:The object of the article is the pore space in asphalt concrete, it is suggested that the title be changed from asphalt to asphalt concrete

We also conducted the tests on porous asphalt. For this reason, we consider the title with asphalt to be appropriate, since two different asphalts were analyzed.

2:The abstract includes too much background information and lacks clear conclusions.

We have revised the abstract so that it fits better to the content of the paper and gives a short overview of the paper. The changes can be seen in the edited submission.

3:The citation format of the references in the text should be revised according to the requirements of the journal

We have adjusted the citation style according to the specifications.

4:What are the technical advantages of using 2D petrographic, given that CT is also capable of obtaining 2D individual sections of concrete?

The technical advantage of 2D petrography is the fast processing of specimens. In addition, very large specimens can be analyzed very quickly. We have included this in the summary (see line 411-412).

Reviewer 4 Report

The research deals with determining air void content and aggregate shape in two type of asphalt mixtures using 3D and 2D scanning methods. The research work is good but the organization needs a little modification. The abstract does not sound like an abstract. The authors tale about roads and importance of air voids and forget to unravel the methodology and especially the main findings of their research. Within introduction, a good literature review is given but the necessity of the current research work in liaison to the literature review is missing. What is missing in the previous studies that this study covers? 
Also, it is important to measure air void based on conventional method because the basics are the criteria. 

Author Response

Dear Reviewer,

Thank you very much for your constructive comments on our paper. We have revised the paper accordingly and tried to fulfill the comments.

In the following, we will briefly explain how we have incorporated your comments and where you can find the changes in the paper.

The research deals with determining air void content and aggregate shape in two type of asphalt mixtures using 3D and 2D scanning methods. The research work is good but the organization needs a little modification. The abstract does not sound like an abstract. The authors tale about roads and importance of air voids and forget to unravel the methodology and especially the main findings of their research. Within introduction, a good literature review is given but the necessity of the current research work in liaison to the literature review is missing. What is missing in the previous studies that this study covers? 
Also, it is important to measure air void based on conventional method because the basics are the criteria.

Thank you very much for the very helpful comments. We have revised the abstract to better fit our paper. You can see the change in the new upload (line 11-29).

In the literature study, we made a change in the last section to indicate why we did the study and what purpose the literature study presented up to that point provides (line 101-108)

The conventional method for determining the void content of asphalt samples is to weigh the samples when dry, under water, and after storage in a water bath. However, this method is very prone to errors and even simple deviations in the weighing results lead to a different void content. For this reason, we looked for methods to obtain a better and more reliable result. The two selected methods are presented and compared in this paper.

Round 2

Reviewer 3 Report

The content of manuscript is acceptable for MATERIALS.